# Charting the spatial dynamics of early SARS-CoV-2 transmission in Washington state

**Tobias S. Brett** [1]*, **Shweta Bansal**[2], **Pejman Rohani**[1,3,4]

**1** Odum School of Ecology, University of Georgia, Athens, Georgia, United States of America, **2** Department of Biology, Georgetown University, Washington, D.C., United States of America, **3** Department of Infectious Diseases, College of Veterinary Medicine, University of Georgia, Athens, Georgia, United States of America, **4** Center for Influenza Disease & Emergence Research (CIDER), Athens, Georgia, United States of America

* tsbrett@uga.edu

**Data Availability Statement:** All data and code used to produce the results shown are archived on zenodo (DOI: 10.5281/zenodo.7724859).

**Funding:** TSB, SB and PR were supported by the National Institute Of General Medical Sciences of

## Abstract

The spread of SARS-CoV-2 has been geographically uneven. To understand the drivers of this spatial variation in SARS-CoV-2 transmission, in particular the role of stochasticity, we used the early stages of the SARS-CoV-2 invasion in Washington state as a case study. We analysed spatially-resolved COVID-19 epidemiological data using two distinct statistical analyses. The first analysis involved using hierarchical clustering on the matrix of correlations between county-level case report time series to identify geographical patterns in the spread of SARS-CoV-2 across the state. In the second analysis, we used a stochastic transmission model to perform likelihood-based inference on hospitalised cases from five counties in the Puget Sound region. Our clustering analysis identifies five distinct clusters and clear spatial patterning. Four of the clusters correspond to different geographical regions, with the final cluster spanning the state. Our inferential analysis suggests that a high degree of connectivity across the region is necessary for the model to explain the rapid inter-county spread observed early in the pandemic. In addition, our approach allows us to quantify the impact of stochastic events in determining the subsequent epidemic. We find that atypically rapid transmission during January and February 2020 is necessary to explain the observed epidemic trajectories in King and Snohomish counties, demonstrating a persisting impact of stochastic events. Our results highlight the limited utility of epidemiological measures calculated over broad spatial scales. Furthermore, our results make clear the challenges with predicting epidemic spread within spatially extensive metropolitan areas, and indicate the need for high-resolution mobility and epidemiological data.

## Author summary

Geographic structure in human populations has been widely recognised as a key determinant in the spread of infectious diseases, however the mechanisms underlying it are hard to disentangle. Using two distinct statistical analyses, we sought to address questions surrounding the impact of movement patterns and stochastic events on the spatial spread of SARS-CoV-2 in Washington state. Through our first analysis, which made use of a clustering algorithm, we uncovered a relationship between spatial proximity and the similarity

the National Institutes of Health under Award Number R01GM123007. PR was supported by federal funds from the National Institute of Allergy and Infectious Diseases, National Institutes of Health, Department of Health and Human Services, under Contract No. 75N93021C00018 (NIAID Centers of Excellence for Influenza Research and Response, CEIRR). The content is solely the responsibility of the authors and does not necessarily represent the official views of the National Institutes of Health. The funders had no role in study design, data collection and analysis, decision to publish, or preparation of the manuscript.

**Competing interests:** None.

of epidemic trajectories. Clusters of similar trajectories formed a clear geographical pattern. In our second analysis, we performed statistical inference on time series data from the Puget Sound region of Washington state. From our inference, we found that both a high degree of connectivity and atypically fast transmission within the first few weeks of the outbreak were necessary to explain the rapid inter-county spread. Furthermore, this stochastic early spread had a lasting effect on the subsequent epidemic course as non-pharmaceutical interventions were insufficient to temporarily eradicate the virus from the region.

## Introduction

SARS-CoV-2, the viral cause of COVID-19, was first detected in samples from Wuhan, China, in late December 2019 [1, 2]. Before travel restrictions were enacted, air travel led to the virus being rapidly dispersed globally [3]. The first confirmed SARS-CoV-2 infection in the US was on January 15 2020 in Snohomish county, Washington state [4], less than three weeks after its reported discovery in Wuhan, prompting intensive contact tracing efforts [5]. A lack of community testing meant that sustained transmission was not suspected until an outbreak in a long-term care facility was detected in neighbouring King county on February 28 [6]. Phylogenetic analysis suggests that cryptic community transmission had been ongoing in Washington state since late January, likely due to a separate introduction from China [7], and that cases were substantially under-reported [8]. In response, through March 2020 there was an increase in non-pharmaceutical interventions including work-from home orders, mandated closures of dine-in restaurants and recreational establishments and a prohibition of large gatherings [9]. On the state-level, confirmed cases in Washington peaked at the start of April, followed by a smaller summer wave and then a large winter wave (Fig 1A). After the introduction of mass adult vaccination in spring 2021 there have been further waves, including ones associated with each of the alpha, delta and omicron variants of concern (VOC). Geographically, there is substantial separation between populations, which are divided into 39 administrative counties (Fig 1B). On the county-level, we see extensive variation among epidemic trajectories (Fig 1C).

The importance of spatial population structure for determining the transmission dynamics of infectious diseases has been widely recognised for many human diseases, including dengue [10], pertussis [11–13], measles [14], influenza [15, 16], enteroviruses [17, 18] and now SARS-CoV-2 [19–23]. A range of mechanisms can underlie spatial variation, including: the relative transmissibility of locally dominant strains [24], the timing of virus introduction [25], spatial connectivity network topology [26], drivers of human mobility [13, 27], population demography [28, 29], variation in non-pharmaceutical interventions [27], levels of population immunity (both natural- and vaccine-derived) [30] and differences in climate [17, 18, 31]. Studies using spatially-explicit transmission models have been performed to understand facets of the COVID-19 pandemic, including quantifying the impact of vaccination and the emergence of novel variants on disease burden [19], reconstructing undocumented infections and their impact on estimates of infection fatality ratios [20, 21], and understanding the invasion dynamics of novel variants [22] and their transmission advantage over previously circulating strains [23].

In this study we intended to understand how stochasticity in the transmission process combined with commuting network structure to shape the initial phases of the SARS-CoV-2 epidemic. Furthermore, we sought to do this in the face of time-varying trends in case ascertainment and localised variation in the rollout of public health intervention measures and

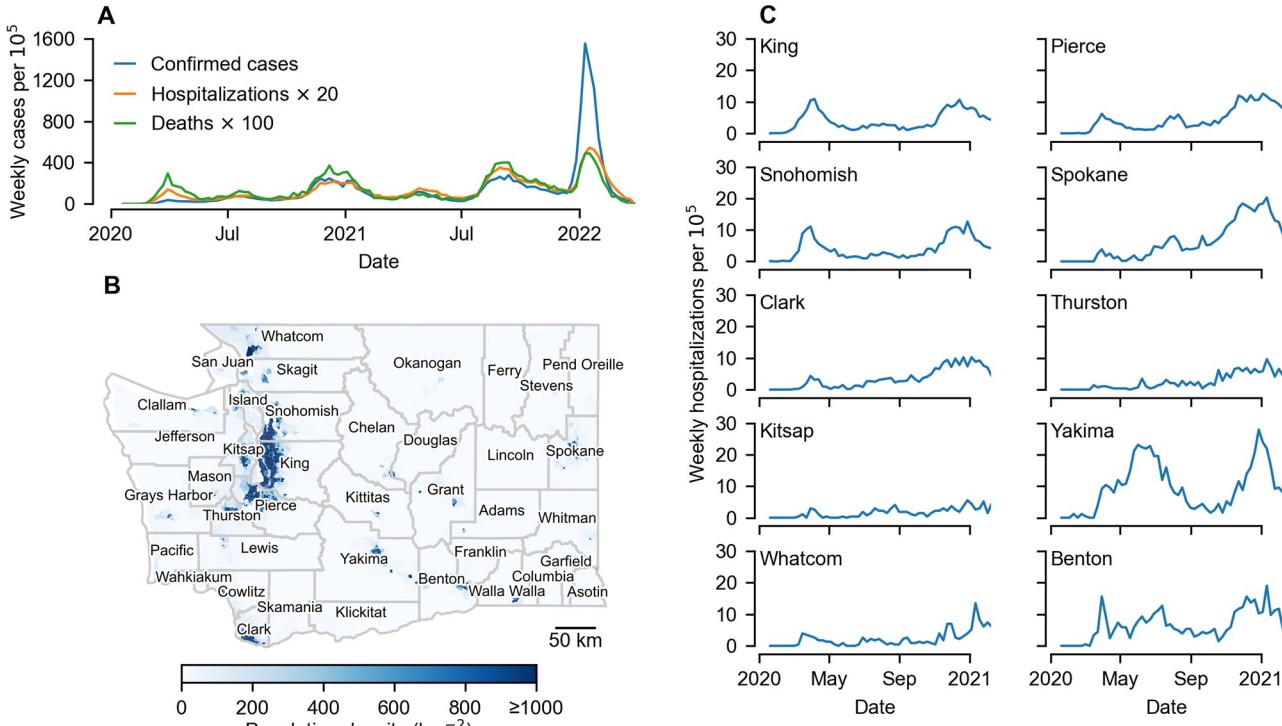

**Fig 1. Early dynamics of the COVID-19 pandemic in WA.** A) Broadly, the spread of SARS-CoV-2 can be characterised into successive waves—three prior to the introduction of mass vaccination in early 2021, and then three post-vaccination waves each associated with a variant of concern: first alpha, then delta, then omicron. The changes in the relative peak sizes of weekly confirmed cases, newly hospitalized cases and deaths highlight some of issues with uncovering SARS-CoV-2 spread. In particular, we draw attention to the relatively small number of confirmed cases during the first wave, likely due to limited testing capacity. B) Further complicating understanding spread is the spatial distribution of the population. Around 70% of the population is concentrated in the Puget Sound region (King, Pierce, Snohomish, Thurston and Kitsap counties) with other population clusters including the Tri-cities area and Yakima valley (Benton, Franklin and Yakima counties) and the cities of Spokane, Bellingham and Vancouver (in Spokane, Whatcom and Clark counties respectively). C) Focusing on the pre-vaccination period, time series of county-level hospitalizations exhibit clear variation across counties. Map makes use of TIGER/Line Shapefiles from the U.S. Census Bureau (https://www.census.gov/geographies/mapping-files.html).

compliance (e.g. extent of social distancing). To accomplish this, we interrogated time series data from Washington state using two statistical analyses. We first used a clustering analysis to examine spatial patterns, uncovering clear geographic structure to the spread of SARS-CoV-2. This highlights the potential limitations in using state-level incidence data to infer epidemiological mechanisms. We subsequently used a stochastic spatial transmission model to perform likelihood-based inference on hospitalizations time series data from five counties in the Puget Sound region (belonging to two clusters in the preceding analysis and including the Seattle metropolitan area). Maximisation of the likelihood and uncertainty quantification was achieved using a refinement of the iterated filtering algorithm [32, 33]. We found that the data are most consistent with a high degree of connectivity across the region. Concordant with the findings of phylogenetic studies [7], our results suggest that a single introduction was responsible for much of the early outbreak. However, to achieve the outbreak size observed in March and April our inference favours an earlier time of introduction in mid-January (contemporaneous with the first identified case), rather than at the start of February. Furthermore, our results suggest that the size of the outbreak in Snohomish and King counties required atypically rapid spread (both inter- and intra-county) within the first couple of weeks after introduction.

Our results highlight that when an introduction escapes detection and successfully seeds an outbreak, the resulting outbreak can be substantially larger and more widespread than predicted by deterministic models of epidemic growth. Together, our results testify to the lasting impacts of geography and stochasticity on the spread of SARS-CoV-2, with ramifications for on-going attempts at forecasting epidemic trajectories—in particular within densely populated urban areas.

## Results

### State-wide clustering analysis of epidemiological time series

To understand which counties in Washington state experienced similar epidemic trajectories, and whether these similarities were related to spatial proximity, we performed a clustering analysis of the rank correlation (quantified using Spearman's $\rho$) among county-level time series of weekly confirmed cases per $10^5$ (Fig 2; see S1 and S2 Figs for the underlying time series data). Three sparsely populated counties, Garfield, Skamania and Wahkiakum, with populations under 5,000 were excluded due to insufficient data. We found dynamics of COVID-19 were positively correlated between all pairs of counties, with correlation ranging from $\rho = 0.27$ (between Yakima and Whitman) to $\rho = 0.98$ (between Clark and Spokane) (Fig 2A). To explore groupings in the pair-wise correlation between counties, we performed a clustering analysis using hierarchical agglomerative clustering with a maximum linkage criterion (Fig 2B). We found specific closely linked groupings of neighboring counties, including Pierce, Kitsap and Mason, Franklin and Benton, and King and Snohomish. However, other closely linked counties were geographically far apart, in particular Clark, Spokane and Thurston. Based on the silhouette score (a measure of clustering performance [34], see S3 Fig), we detected five clusters (Fig 2B). Mapping these clusters elucidates clear geographic patterns (Fig 2C).

Three of the five clusters were contiguous: the first (yellow) centered around the Puget Sound region includes Seattle (located in King), the second (orange) includes the Yakima valley and its confluence with the Columbia river in the Tri-cities area (located on the border of Franklin and Benton) and the third (teal) in the rural north central region. Of the two remaining clusters, one (blue) is mostly contiguous apart from Ferry, which is the the most distantly linked county in the data set. The final cluster (pink) is the largest, comprising a number of urban and rural areas across the entire state (including the previously mentioned Pierce, Kitsap, Clark, Spokane and Thurston counties).

We repeated the analysis on weekly hospitalizations data (S4 Fig). The data were substantially noisier, due to the smaller numbers of infections that were hospitalized, which is reflected in lower entries in the correlation matrix. Overall, the clusters were less well defined, as quantified using the silhouette score (S5 Fig). As might be expected, the impact of increased noise is most pronounced on the clustering of smaller counties. Apart from Thurston, the top ten most populous counties all remain in the same clusters (S4 Fig).

### Statistical inference using spatio-temporal data from the Puget Sound region

To investigate further the factors driving county-level differences in epidemic trajectories, including the role of human mobility patterns and non-pharmaceutical interventions, we used a stochastic spatial transmission model to perform statistical inference on county-level weekly hospitalizations data. For the analysis, we selected five counties, belonging to two clusters, that form the core of the Puget Sound metropolitan area: King, Pierce, Snohomish, Thurston and

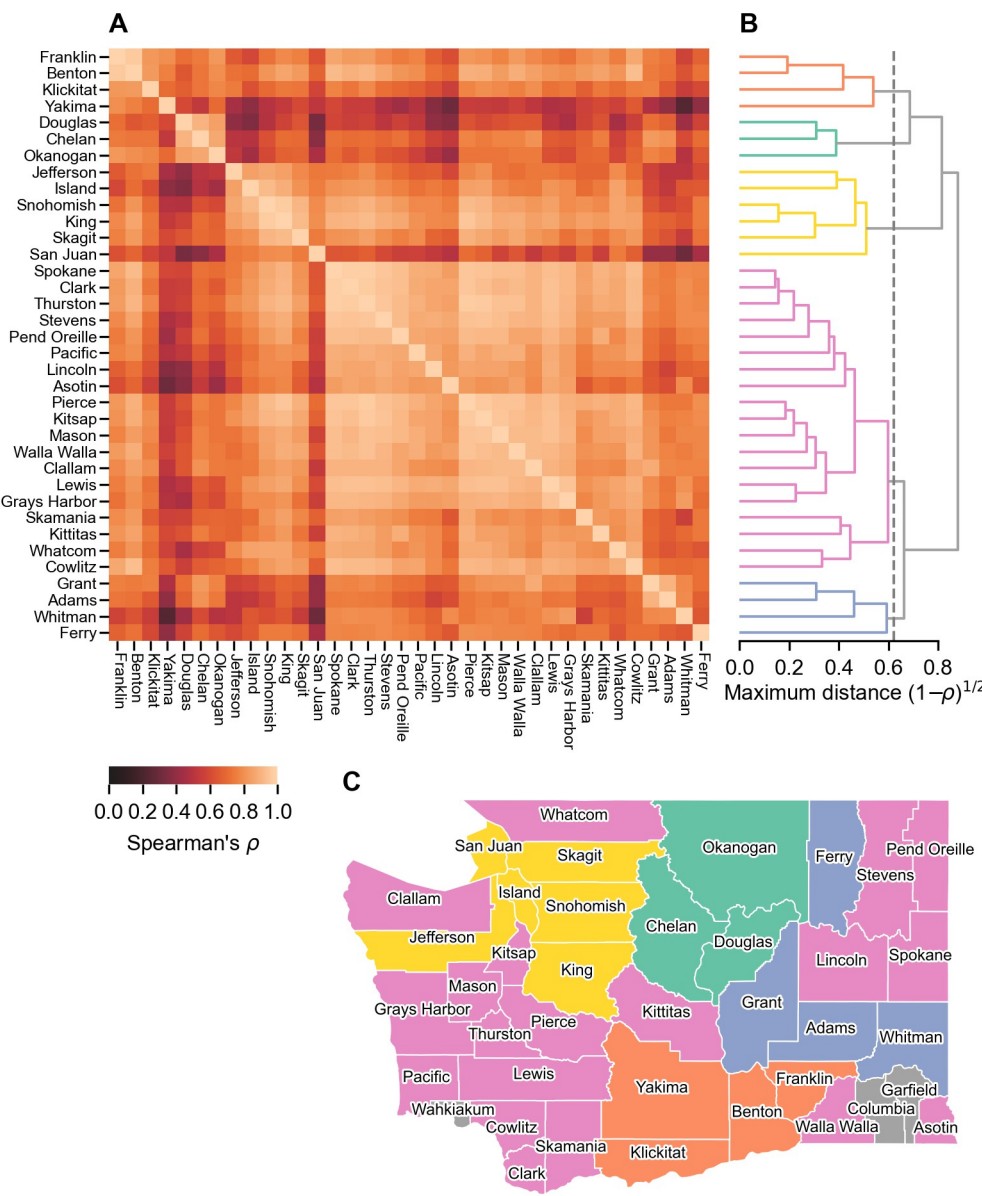

**Fig 2. Analysis of temporal correlation in pre-vaccination incidence time series in WA state.** A) Correlation matrix of Spearman's $\rho$ for each pairing of WA counties. Three sparsely populated counties (Garfield, Skamania and Wahkiakum) were excluded from the analysis due to insufficient data. B) Results of agglomerative clustering performed on the correlation matrix. Based on the silhouette score, we identify five distinct epidemiological clusters. C) Map of identified clusters. We observe a clear spatial structure to the clustering: i) a cluster predominantly of counties from Seattle northwards bordering the Puget Sound (yellow), ii) a cluster including the Yakima valley and Tri-cities area (orange), iii) two rural clusters in eastern Washington (teal and blue) and iv) a state-wide cluster (pink). Map makes use of TIGER/Line Shapefiles from the U.S. Census Bureau (https://www.census.gov/geographies/mapping-files.html).

Kitsap. Together these counties account for over 70% of the total state population. We focused on hospitalization data, rather than confirmed cases, to alleviate issues with time-varying testing rates (see S6 Fig). To allow us to focus solely on questions around the impact of movement patterns and local non-pharmaceutical interventions on the spatial spread of SARS-CoV-2 in Washington state, we restricted our analysis to the dynamics of SARS-CoV-2 up to February

2021, prior to both i) the introduction of mass vaccination and ii) the introduction of VOCs in 2021 (with phylogenetic evidence suggesting that in 2020 co-circulating strains in Washington had comparable reproductive numbers [35]). We further filtered the data to include only hospitalized cases aged under 60, due to sustained transients in older age groups (possibly due to their increased ability to socially distance compared to school- and work-aged individuals, see S6 Fig). To model the impact of county-specific non-pharmaceutical interventions, for each county we introduced a resident reproductive number, $R_i^{\text{res}}$, defined as the expected number of cases caused by a non-commuting individual who remains in county $i$ with a fully susceptible population. This function was parameterised for each county via a set of basis function coefficients, which were estimated from incidence data (see Methods). In addition, we also obtained maximum likelihood estimates for four region-wide parameters: the baseline reproductive number ($R_0$), the probability an infection was aged under 60 and hospitalized ($\rho$), the average number of weekly cases imported to the region ($\eta$) and the baseline pre-social distancing fraction of total contacts a commuting individual would have in their commuting destination ($\epsilon$). Maximising the log-likelihood was accomplished in two steps (see Methods). First, for each point on a grid of region-wide parameter values we used iterated filtering to maximise the log-likelihood over the space of basis function parameters (S7 Fig). This likelihood surface shows a trade-off in model performance between $R_0$, $\rho$ and $\eta$, which we attribute to all three parameters affecting the apparent size of a growing epidemic. The fraction of contacts in the commuting destination, $\epsilon$, does not trade off with any of the other region-wide parameters. Maximum likelihood estimates for all parameters were subsequently obtained by maximising the likelihood over the grid, with profile likelihoods and 95% confidence intervals (estimated using Wilks' theorem [36]) for each region-wide parameter shown in Fig 3A–3D.

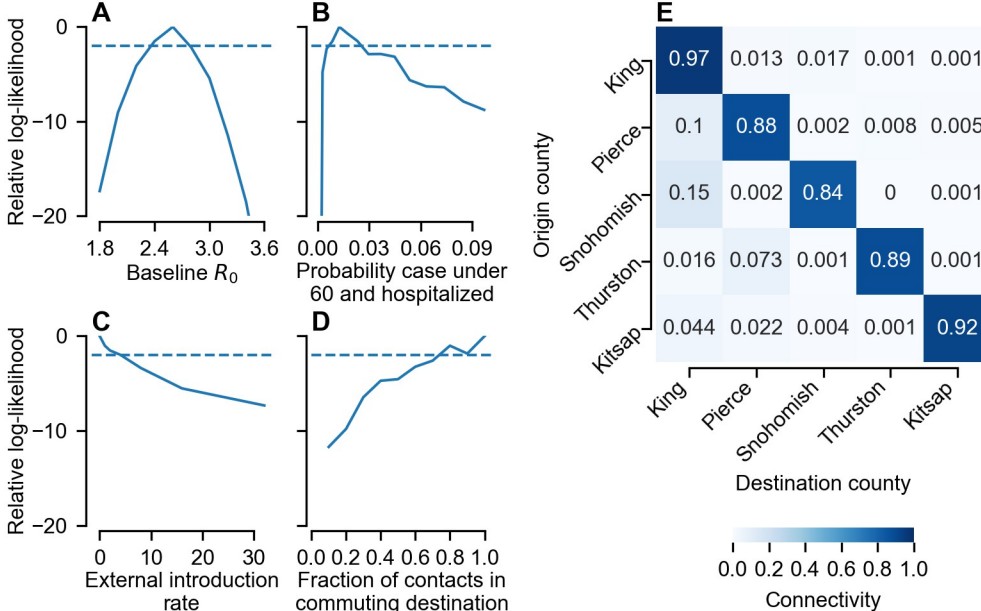

**Fig 3. Profile likelihoods and reconstructed connectivity matrix from the inference analysis.** A–D) Profile likelihoods are shown for the four estimated global parameters: the baseline reproductive number (A), the probability an infection is under 60 and hospitalized (B), the weekly rate infections are imported into the population assuming full susceptibility (C) and the pre-pandemic fraction of contacts that commuting individuals have in their commuting destination (D). The dashed horizontal line corresponds to 2 log-likelihoods below the maximum. E) Reconstructed connectivity matrix (see Methods) incorporating census commuting data and the maximum likelihood estimate of the commuting parameter $\epsilon$ (see panel D).

A sensitive question in this inference is the start date of the epidemic. In the analysis above, we assumed a starting date of January 12 (the date of the first reported case in Washington state). However, a phylogenetic study by Worobey et al. [7] suggested the outbreak was initiated during the week beginning January 26 (the median of their posterior distribution), we therefore examined the sensitivity of our statistical inference to initializing on this date. We found the model performance (considering only the data post-January 26) was 7.7 log-likelihoods lower compared with an outbreak starting in the week of January 12. In addition, with the later start date, the maximum likelihood estimate for $\eta$ suggests an average of 8 introductions per week, with $\eta = 0$ lying outside the 95% confidence interval, making it extremely improbable that all infections in the outbreak were descended from a single introduction (as suggested by the results of Worobey et al. [7]). This is in contrast with our inference using the earlier January 12 start date, where the MLE was $\eta = 0$ (Fig 3C).

Using the demographic data, commuting data and our maximum likelihood estimate for $\epsilon$, we reconstructed the connectivity matrix: the proportion of daily contacts an individual resident in the origin county has with individuals in the destination county. We found that the reconstructed connectivity matrix is the same regardless of whether a January 12 or January 26 start date was used (Fig 3E). At the beginning of the pandemic (before the impact of local variation in $R_i^{\mathrm{res}}$), this also gives the proportion of transmission in each destination county by an infectious individual resident in the origin.

As explained above, $R_i^{\mathrm{res}}$ is the reproductive number of individuals who do not travel outside of their county of residency. The reproductive number for commuters is a weighted sum of $R_i^{\mathrm{res}}$ in their origin and commuting destination. Time-variation in $R_i^{\mathrm{res}}$ alters the rate of transmission between individuals located in a county–either due to residency or commuting (Fig 4). We find that non-pharmaceutical interventions sharply reduced $R_i^{\mathrm{res}}$ in all five counties, with a particularly sharp drop in Thurston (Fig 4I) and Kitsap (Fig 4J). Crucially, while non-pharmaceutical interventions drove $R_i^{\mathrm{res}}$ below 1, they were of insufficient extent and duration to eliminate SARS-CoV-2 in the Puget Sound region. Subsequent waves in the summer and winter were driven by time-varying trends in $R_i^{\mathrm{res}}$.

## Impact of stochasticity on the epidemic trajectory in the Puget Sound region

Our maximum likelihood estimate (MLE) for the commuting parameter was $\epsilon = 1$, i.e. the strongest connectivity possible (see Fig 3D). To understand why the model favoured high spatial connectivity, we systematically varied $\epsilon$ and investigated the log-likelihood of each week's incidence conditioned on the preceding data, i.e. $\ln P(x_t|\{x_s\}_{s<t})$, which we refer to as the conditional log-likelihood (CLL; Fig 5A). The CLL paints a picture of how well the model explains each data point and helps identify those that contribute most to the overall log-likelihood, which is the sum of the CLL across all weeks. We find that the CLL follows a similar trajectory regardless of the value of $\epsilon$, and is generally lowest when the epidemic is largest, an expected consequence of both demographic stochasticity [37] and observation error (see Eq 10). More insight can be gained into the dependence of the CLL on $\epsilon$ by calculating the relative CLL, found by subtracting the CLL corresponding to the MLE $\epsilon = 1$ (Fig 5B). We see that lower values of $\epsilon$ (darker colors) are penalised the most early in the epidemic (pre-April 2020), and have the lowest relative CLL. During April and May, there is a suggestion that smaller values of $\epsilon$ have a larger CLL than the MLE (relative CLL > 0). To quantify these observations further, we measure the association between the CLL and $\epsilon$ for each week using Kendall's $\tau$ (Fig 5C). Pre-March 2020 lockdown, we see the largest values of $\tau$, implying the strongest positive association between $\epsilon$ and CLL. After the start of lockdown, Kendall's $\tau$ drops. Indeed, during late

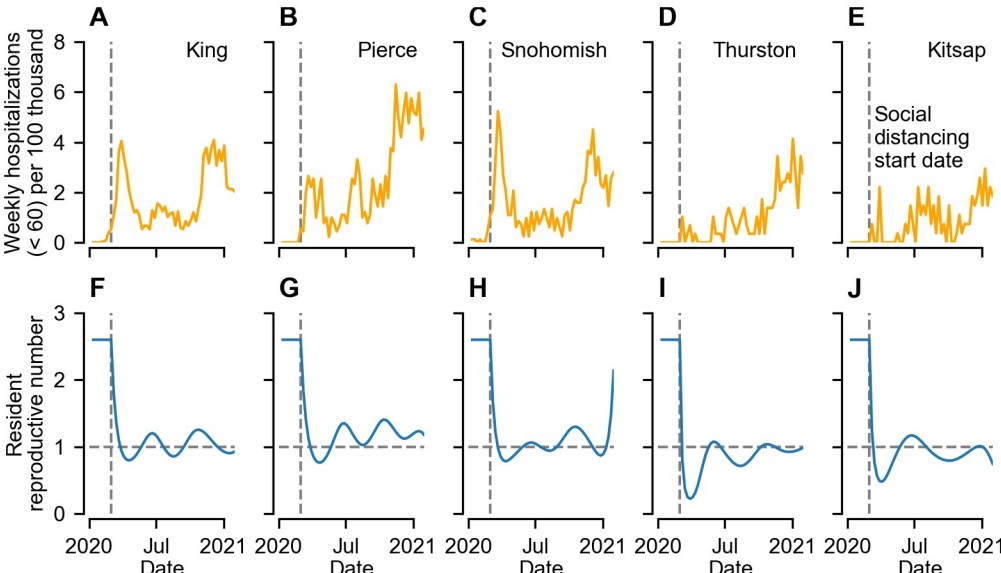

**Fig 4. Time series of weekly COVID-19 hospitalizations and maximum likelihood estimates of the resident reproductive number for the five counties studied in inference analysis.** A–E) Weekly hospitalized cases under the age of 60 for King (A), Pierce (B), Snohomish (C), Thurston (D) and Kitsap (E). Inference was performed on hospitalizations data rather than confirmed cases due to temporal variation in testing access (see main text). F–J) Reconstructed maximum-likelihood estimates of the resident reproductive number for each county (see Methods for details). Time-varying trends reflect the combined impact of all non-pharmaceutical interventions (including social distancing and mask wearing). Non-pharmaceutical interventions were assumed to start in the week beginning March 1 2020, i.e. immediately after the first discovery of sustained SARS-CoV-2 transmission in the state.

April and May, we see a period where $\tau < 0$, implying a negative association between $\epsilon$ and CLL. From June onwards $\tau$ stabilises, fluctuating about a smaller positive value than pre-lockdown, indicating a weaker positive association. Overall, we see that the main reason the model parameterized with higher values of $\epsilon$ performs better is due to an increased ability to fit the data pre-April 2020.

In addition to investigating the CLL, we also investigated the importance of each weekly hospitalisations data point in determining the subsequent course of the regional epidemic (Fig 5D). This was quantified via the log-probability of the final week's hospitalizations data (week beginning January 31 2021), $x_T$, conditioned on all data up-to-and-including week $t$, $Q_t = \ln P(x_T | \{x_s\}_{s \leq t})$. An increase (decrease) in $Q_t$ in week $t$ implies that the new observation $x_t$ increases (decreases) the probability of observing $x_T$ at the later time $T$.

We see the largest rise in $Q_t$ during March and April, implying that the impact of early stochastic events persisted over the entire period of study. Subsequently, there is little increase in $Q_t$ between April and October 2020, implying that individual stochastic fluctuations which occurred during this second period had limited lasting impact. Finally, from around October onward we see an accelerating rise in $Q_t$. This observation is to be expected, and is analogous to the time series autocorrelation rising as the lag between observations shrinks [37]. Overall we find $Q_t$ is positively associated with $\epsilon$, as expected given our results for the CLL (Fig 5A–5C).

To investigate the dynamical impacts of these early stochastic events, we simulated 50,000 realizations of our fitted transmission model (Fig 6). We find that if the simulations are initialized in the week beginning January 12 2020 (when the first case of COVID-19 was detected in Washington state), the mean trajectory (calculated by averaging over all realizations) predicts

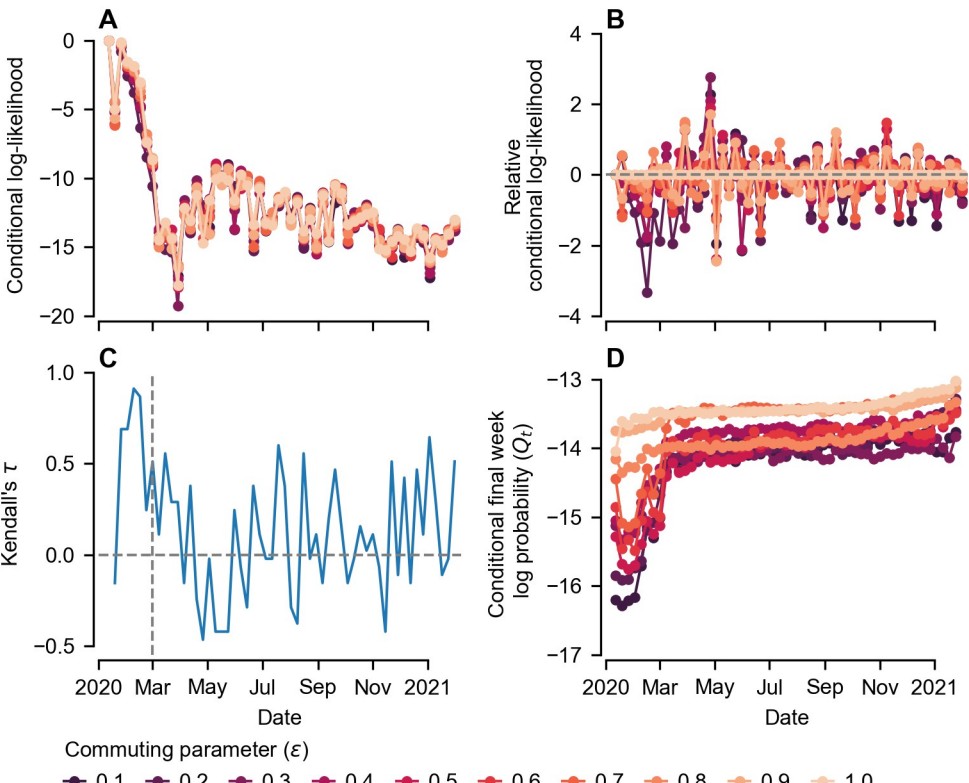

**Fig 5. Investigation of the dependence of model performance on the commuting parameter, $\epsilon$.** A) Conditional log-likelihood for each week of data. The model parameters for each line maximise the overall log-likelihood with $\epsilon$ constrained to the value indicated in the legend. Consequently, summing over all weeks for a particular value of $\epsilon$ gives the corresponding value of the profile log-likelihood shown in Fig 3D. B) Conditional log-likelihood relative to the maximum ($\epsilon = 1$). C) Kendall's $\tau$ coefficient for each week, quantifying the association between conditional log-likelihood and $\epsilon$. A more positive (negative) $\tau$ indicates a stronger positive (negative) association between $\epsilon$ and the conditional log-likelihood for that week's data. D) Log-probability of the final week's data (week beginning January 31 2021) conditioned on all data up to and including the week indicated ($Q_t$). Results show a large jump in $Q_t$ over the course of the first month, indicating the importance of atypical spread in disseminating the virus across the region.

lower hospitalizations than observed throughout the epidemic (Fig 6A). The most similar realization (as quantified via sum of squared errors) tracks the data closely, however falls in the tail of the distribution, suggesting it is an atypical trajectory. If, instead, we simulate from the model using the state of the most similar trajectory in the week of February 9 (i.e. 4 weeks after the first confirmed case) as the initial condition, we find that the sample mean overlaps the data, and the variance among simulated realizations is much smaller (Fig 6B). This substantiates our earlier observations regarding the pronounced early increase in $Q_t$ (Fig 5D). Our results suggest that rapid, atypically intensive transmission within the first couple of weeks after introduction is necessary to generate the epidemic trajectories observed in King and Snohomish counties. Subsequently, the epidemic is sufficiently large that it followed a roughly deterministic trajectory determined by the combined effects of non-pharmaceutical interventions and susceptible depletion.

## Model validation

Finally, from our simulated trajectories at the MLE initialised on February 9 2020 (Fig 6B), we reconstructed the cumulative prevalence of SARS-CoV-2 infection in Washington state

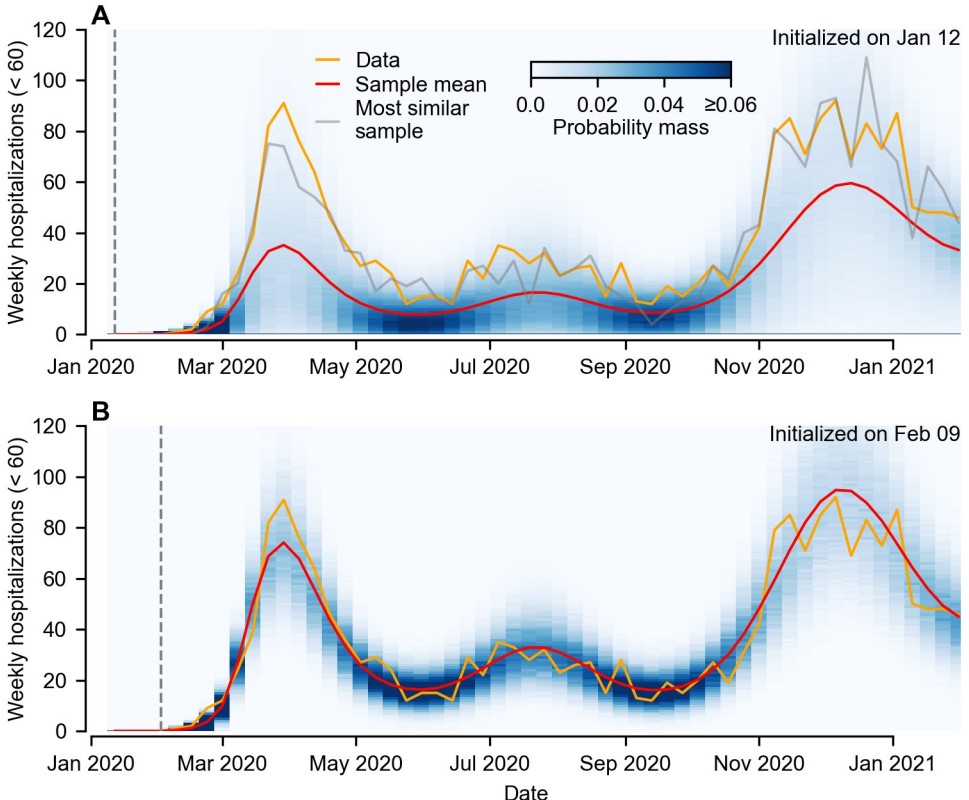

**Fig 6. Comparison of simulated trajectories using the maximum likelihood parameter estimates with observed data.** A) Simulated trajectories for King county initialized on January 12 2020. Blue shading corresponds to the probability mass estimated from 50,000 sampled trajectories, with the sample mean shown in red. The orange curve is the observed hospitalizations data used for parameter inference, and lies in the tail of the probability distribution—suggesting an atypical trajectory. Also shown is the simulated trajectory most similar to the observed data (grey curve) as calculated via the sum of errors. B) Simulated trajectories for King county initialized on February 9 2020 using the state variables of the most similar sample (see panel A). Our results suggest that, once the initial atypical stochastic spread had occurred, the subsequent epidemic trajectory was largely a deterministic response to non-pharmaceutical interventions and susceptible depletion.

through time (Fig 7). We compared our reconstruction with those from four published studies [38–41]. We also compared our simulations with estimates of SARS-CoV-2 antibody seroprevalence released by the CDC [42, 43]. We find our simulations are broadly consistent with two studies (Davis et al. [38] and Wu et al. [41]), and are smaller than the estimates of Pei et al. [39] and Reese et al. [40] by a factor about 1.5 and 3 respectively.

## Discussion

In this study we sought to understand the combined effects of stochasticity and spatial connectivity on the initial spread of SARS-CoV-2 in Washington state. Spatial heterogeneity in the spread of SARS-CoV-2 has been observed elsewhere [19–23, 38], and, as our results suggest, Washington state is no exception. Our findings reveal substantial regional variation in the progress of the epidemic across the state, even prior to the geographically uneven uptake of vaccines (Fig 2). Through our study of the metropolitan Puget Sound region, we find that high connectivity (Fig 3D) coupled with rapid early stochastic transmission(Figs 5 and 6) is necessary to explain the inter-county spread of the virus. In combination, our results highlight the

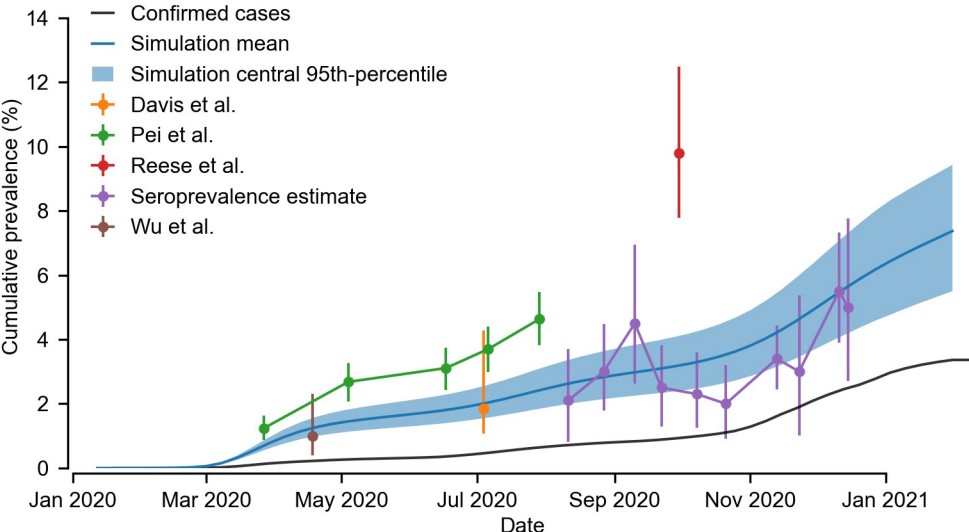

**Fig 7. Comparison of reconstructed prevalence with estimates from the literature.** Using our simulated trajectories initialised on February 9 2020 (see Fig 6B), we estimated the cumulative percentage of the population exposed to SARS-CoV-2 in the Puget Sound region (blue). We compared our estimate to a) time series data of confirmed cases in the Puget sound region (black) b) estimates from four studies (orange, green, red, brown) [38–41] and c) estimates of the seroprevalence in Washington state from serological data (purple) [42, 43]. Caution must be taken in reading these results as the studies used different statistical approaches and spatial aggregations. Davis et al., Pei et al. and Wu et al. all used Bayesian approaches, results shown are the median and the 90% (Davis et al.) and 95% (Pei et al. and Wu et al.) credible interval. The results shown from Reese et al. and the seroprevalence study are the mean and 95% confidence intervals. Our results show the mean and 95% percentile of simulations, accounting for stochastic variability but not parametric uncertainty (see Fig 3). Furthermore, the results from Pei et al. are for the Western Washington region (which includes the Puget sound region), the results from Davis et al., Wu et al., and the seroprevalence study are for the entire state, and the results of Reese et al. are for the combined states of Alaska, Idaho, Oregon and Washington.

limited utility of epidemiological measures calculated over broad spatial scales, which can conceal emerging transmission hot spots and local variation in reproductive number.

Our clustering analysis is able to identify a "typical" Washington state COVID-19 trajectory in 2020: three temporal waves of successively increasing magnitude in spring, summer and winter (the pink cluster shown in Fig 2). The presence of this "typical" highly-correlated cluster stands in contrast with the "travelling wave" pattern of spatial spread, observed for many human infectious diseases (including dengue [10], pertussis [44], measles [14], and influenza [15]). Instead, this cluster points to a remarkable degree of synchrony in the epidemic trajectories of counties from across the state. For instance, the mid-sized Clark, Spokane and Thurston counties are among the most highly rank correlated (see Fig 2A) and have a near-simultaneous initial epidemic take-off (see S10 Fig) despite their geographic separation. The drivers of this synchrony are likely complex and multifaceted, however we suspect it stems from a combination of i) near-simultaneous seeding of the epidemic across the state (aside from the early seeding of the Seattle metropolitan area) and ii) state-wide imposition of social distancing measures (including a prohibition of large gatherings and restaurant closures in spring and a mask mandate in summer 2020) [9].

Importantly, we identify four clusters of contiguous counties that deviate from this typical trajectory. For instance, the yellow cluster, centered on King and Snohomish counties (together 40% of the state's population) had an atypically early and large initial wave, likely due to a combination of it being seeded directly from China [7, 8] and rapid early transmission (see below). We hypothesize that other factors were at play in the orange Yakima valley cluster,

where the initial non-pharmaceutical interventions in spring 2020 appear to have been less effective at controlling transmission than elsewhere, prolonging the spring wave into the summer. Our clustering analysis highlights how state-level indicators can be misleading, for instance suggesting a growing state-wide epidemic when infections are confined to specific regions, or delaying the identification of transmission hot spots, such as in the Yakima valley during April 2020.

We found that the importance of spatial mobility patterns in determining the Puget Sound epidemic trajectory was most acute during the initial phase of the epidemic, i.e. before the identification of community transmission. By considering the conditional log-likelihood of each weekly data point, we found there was the strongest positive association between model performance and connectivity strength prior to the March 2020 interventions (see Fig 5). During this period, high spatial connectivity provides the highest likelihood of the observed rapid transmission between Snohomish and King counties. Once the epidemic was established across the region, the subsequent county-level trajectories were less sensitive to the value of the commuting parameter, reflecting a weaker association with model performance. Our results suggest that accurate quantification of mobility is most important in contexts where incidence is low.

Our reconstruction of the cumulative prevalence of infection in the Puget Sound region suggests that over the course of 2020 about 7% of the population were exposed to SARS-CoV-2. While our reconstructed prevalence is consistent with two previous studies [38, 41], it also broadly tracks the estimated seroprevalence in Washington state published by the CDC [42]. Given this estimate has not been corrected for seroreversion [43], it likely provides an underestimate of the overall percentage of the population exposed to SARS-CoV-2. Our reconstructed prevalence is also lower that in two other studies [39, 40]. This difference could in part be explained by our use of the MLE for model parameters in our reconstruction. In fitting the data, there is a trade-off in model performance between outbreak size and the parameter $\rho$, the probability an infection was aged under 60 and hospitalized: As $\rho$ decreases a larger epidemic is needed to fit the data. The 95% confidence interval on $\rho$ allows for epidemics that are about 50% larger, which would be broadly consistent with Pei et al., however the estimate of Reese et al., from a study using serological data, is about 300% larger [40]. This discrepancy may stem from their estimate being for the combined states of Alaska, Idaho, Oregon and Washington.

After sustained community transmission in Washington state was first identified at the end of February 2020, our results suggest that behavioural changes and other non-pharmaceutical interventions over the course of March rapidly reduced the resident reproductive number from around 2.6 (an estimate consistent with a number of other studies, e.g. [2, 20]) to below 1 in all of the Puget Sound region counties (see Fig 4). During the period of study, we estimate a limited impact of susceptible depletion on SARS-CoV-2 transmission. Instead, our results suggest that the recurrent outbreaks were driven by trends in human behavioural practices (see e.g. [45]) and non-pharmaceutical interventions such as social distancing and mask mandates. Although our analysis cannot rule out a putative role of seasonality (either in viral transmissibility [46] or human susceptibility to infection [47]), its impact is expected to be small due to the high population-level susceptibility [48].

Our analysis highlights some of the challenges in constraining spatial epidemiological models using census commuting data, a common practice in epidemiological modeling (see e.g. [49–52]). Firstly, the rapid spread of SARS-CoV-2 between Snohomish and King counties may have been facilitated by the mobility patterns of certain key population groupings, for instance healthcare workers, who are not representative of the population average captured in the census data. Secondly, while for most of 2020 we observed that model performance was positively

correlated with connectivity strength, during the period of most intensive social distancing (April—June), this correlation was inverted. Instead, performance was better at smaller values of the commuting parameter. If non-pharmaceutical interventions resulted in a disproportionately large reduction in inter-county transmission compared to intra-county transmission, as was likely the case, this would explain the reversal in model performance. This might also explain the large drops in $R_i^{\text{res}}$ observed in Kitsap and Thurston counties during this period, the two counties with the lowest per capita hospitalizations. Parameterising models using accurate time-varying data on human mobility would help alleviate this issue. Overall, however, our results indicate this second phenomenon was of smaller importance to model performance than the early rapid transmission outlined above.

Uncertainty in data quality has proven a challenge to understanding the early spread of the SARS-CoV-2 pandemic. As we explained above, we chose to perform our inference on time series data of hospitalised cases aged under 60 as they were unaffected by well established time-varying biases, in particular due to test availability issues and improvements in hospitalized patient outcomes [53, 54]. That said, hospitalisation data may be affected by time varying trends in care seeking behaviour (e.g. due to public awareness) or hospital admission criteria (e.g. due to hospital capacity limitations). Unfortunately, we were unable to find documented evidence of the extent and timing of these possible biases.

Phylogenetic evidence suggests that, while the Puget sound outbreak was likely seeded from a single introduction, it was unlikely to stem from the initial introduction in Snohomish county [7]. Instead, Worobey et al. find that the outbreak was likely sparked by separate undetected introduction on February 1 2020 (with a 95% highest posterior density (HPD) of January 14 to February 15) [7]. Our results are not inconsistent with other introductions sparking the outbreak ($\eta \leq 2$ is within the 95% confidence interval, see Fig 3C). After repeating our analysis using an outbreak start date of the week beginning January 26 (S8 and S9 Figs), we find that the maximum log-likelihood of the data post-January 26 is 7.73 log-likelihoods lower compared with using our original start date, the week beginning January 12. Furthermore, $\eta = 0$ is outside the 95% confidence interval, implying that the later start date requires a number of additional introductions in order for the epidemic to be large enough to be consistent with the hospitalizations data. In summary, while our results do not necessarily require the outbreak to be seeded by the Snohomish introduction, in order to explain the number of hospitalizations in March and April the epidemiological data favour a putative undetected introduction occurring at the beginning of the 95% HPD of Worobey et al.

In addition to requiring high spatial connectivity, we also found the outbreaks in both King and Snohomish counties at the start of lockdown were roughly two and three times larger respectively than would be expected from a typical (mean) realization of the best-fitting model (see Fig 6). If the outbreak was seeded from a single later introduction in the week of January 26 (see previous paragraph), the same result occurs (S11 Fig). This observed phenomenon is due to demographic stochasticity in the disease transmission process, and suggests that atypically large bursts of transmission occurred in the first couple of weeks. Dynamically, this is a property of stochastic birth-death processes [55], of which the early phase of an outbreak is an example. In essence, an outbreak sparked in a population that successfully avoids dying out is more likely to have spread faster than average—mathematically, if the probability of (local) extinction is $P(X_t = 0) > 0$ then $\mathbb{E}[X_t | X_t > 0] > \mathbb{E}[X_t]$. The impact of individual high transmission events on avoiding extinction is more pronounced the earlier in the outbreak they occur, and their occurrence has the effect of "fast-fowarding" the epidemic, with the resulting epidemic size being larger than would be expected from the mean trajectory.

Forecasting the spread of infectious diseases is increasingly a goal of infectious disease modellers (see e.g. [56]). Our findings indicate a pronounced impact of geography and stochasticity on the spread of SARS-CoV-2 which can not be ignored in such efforts—understanding the extent to which observed clusters are shaped by intrinsic similarities between neighbouring counties versus their proximity within the wider spatial transmission network is a pressing topic for further study. Finally, our results demonstrate that accurate prediction of future spread within densely populated urban areas during periods of low incidence is not feasible without, at a minimum, i) modelling approaches capable of accounting for stochastic transmission events and ii) access to zip-code- or neighbourhood-level epidemiological and mobility data.

## Methods

### Data sources

Throughout the pandemic, the Washington state Department of Public Health has released weekly COVID-19 data, consisting of confirmed cases, new hospitalizations and deaths. Data are disaggregated on the county-level and by age [57]. As our study focused on understanding spatial patterns in SARS-CoV-2 transmission, we restricted our analysis to data from the period January 12 2020 (the week of the first confirmed case) to January 30 2021. This end date was chosen to be prior to the introduction of novel VOC and mass vaccination, both of which have a confounding spatially heterogeneous effect on viral transmission.

For our likelihood-based inference we used data on the weekly number of new COVID-19 hospitalizations, in order to avoid confounding time-varying trends in confirmed case reports and mortality data (due to the limited initial testing capacity and improvements in treatment respectively). In addition, we excluded hospitalizations in individuals over 60 from the data due to early outbreaks in residential care homes [6].

Demographic data on the population size of each Washington county were sourced from the US census [58]. Mobility data recording the number of daily commuters between each county in Washington state were also sourced from the US census [59]. State-level seroprevalence estimates of cumulative SARS-CoV-2 infections were periodically made publicly available by the CDC [42, 43].

### Clustering analysis

To understand state-wide patterns in SARS-CoV-2 transmission, we performed a clustering analysis on time series data [60]. We used county-level confirmed case data, excluding Garfield, Skamania and Wahkiakum counties (each with population <5000) from the analysis due to insufficient data. In the clustering analysis, we used $1 - \rho$ as our distance matrix, where $\rho$ is the matrix of pairwise values of Spearman's $\rho$ calculated from the county-level confirmed case time series. We used hierarchical agglomerative clustering with a maximum linkage criterion to identify clusters of counties [61]. The number of clusters present in the data was determined by maximising the silhouette score [34]. We repeated our analysis using hospitalization data in place of case counts.

### Transmission model

Statistical inference was performed using a stochastic *SEIR*-based transmission model that modelled transmission of SARS-CoV-2 in a population of $n_c$ counties. As each county had distinct transmission parameters (see below), the curse of dimensionality limited the computational feasibility of our statistical inference to $n_c = 5$. We chose to focus on the five counties

which form the core of the Puget Sound region: King, Pierce, Snohomish, Thurston and Kitsap. We denote the number of new infections in week $[t, t+1)$ among residents of county $i$ by $I_{i,t+1}$. Assuming a 7-day serial interval (consistent with multiple estimates for the original 2020 variant [2, 62, 63]), $I_{i,t+1}$ can be modelled using a Poisson distribution, with expectation

$$\mathbb{E}[I_{i,t+1}] = \lambda_{i,t} S_{i,t}, \tag{1}$$

where $\lambda_{i,t}$ is the force of infection and $S_{i,t}$ is the number of susceptible individuals resident in $i$ at time $t$. We update $S_{i,t}$ according to the update rule

$$
\begin{aligned}
S_{i,t+1} &= S_{i,t} - I_{i,t+1} \\
&= S_{i,0} - \sum_{t'=1}^{t+1} I_{i,t'}.
\end{aligned}
\tag{2}
$$

Due to a lack of evidence to the contrary, we assumed the entire population had no prior immunity against SARS-CoV-2, i.e. that $S_{i,0} = N_i$ where $N_i$ is the population of county $i$.

The force of infection $\lambda_{i,t}$ depends on the contact rates of susceptible individuals resident in location $i$ with infectious individuals, and is given by

$$\lambda_{i,t} = \xi_t \left( \sum_{j=1}^{n_c} \beta_{i,j,t} I_{j,t} + R_i^{\text{res}}(t) \eta / N_i \right), \tag{3}$$

where $\eta$ is the average weekly number of SARS-CoV-2 infections imported into the region and $\xi_t$ is a zero-mean gamma noise process with standard deviation $\sigma_{\text{SE}} = 0.126$ that accounts for environmental stochasticity in the transmission process [64]. The parameter $R_{i,t}^{\text{res}}$ is the resident reproductive number for location $i$, which we detail below. The tensor element $\beta_{i,j,t}$ is the transmission rate from infectious individuals in $j$ to susceptible individuals in $i$ at time $t$,

$$\beta_{i,j,t} = \sum_{k=1}^{n_c} R_{k,t}^{\text{res}} \frac{K_{k,i} K_{k,j}}{\sum_{\ell=1}^{n_c} K_{k,\ell} N_\ell}, \tag{4}$$

where the weight $\frac{K_{k,i} K_{k,j}}{\sum_\ell K_{k,\ell} N_\ell}$ is determined by the connectivity matrix, whose elements $K_{i,j}$ correspond to the probability that an individual resident in $j$ is present in $i$. The denominator $\sum_{\ell=1}^{n_c} K_{k,\ell} N_\ell$ corresponds to the mobility-adjusted population size of location $k$. The sum over $k$ in Eq 4 accounts for the small but non-zero probability that an individual resident in $i$ might encounter an individual resident in $j$ in a third county, $k$. The connectivity matrix was parameterised using commuting data, and has the structure

$$
K_{i,j} =
\begin{cases}
\epsilon \frac{T_{i,j}}{N_j} & \text{if } i \neq j \\
1 - \sum_{i=1; i \neq j}^{n_c} K_{i,j} & \text{otherwise,}
\end{cases}
\tag{5}
$$

where $T_{i,j}$ is the number of daily commuters from $j$ to $i$ in the commuting data and the commuting parameter $\epsilon$, which can take values in $[0, 1]$, is the proportion of contacts that a commuter has in their commuting destination. The diagonal elements of the contact matrix account for both the contacts of non-commuting individuals and commuters in their residence location (i.e. while not at work).

The resident reproductive number for location $i$, $R_{i,t}^{\text{res}}$, is the expected number of secondary infections caused in a fully susceptible population by a non-commuting individual who lives in $i$. This parameter varies through time due to the effects of non-pharmaceutical infections,

such as social distancing and mask wearing, and is county-specific to allow for differences in measure implementation and adherence. We model $R_i^{\text{res}}(t)$ using a set of $n_\alpha$ basis functions, $\{c_\mu(t)\}_{\mu=1}^{n_\alpha}$, via the relationship

$$R_i^{\text{res}}(t) = R_0 e^{\sum_{\mu=1}^{n_\alpha} \alpha_{i,\mu} c_\mu(t)}. \tag{6}$$

The functions $c_\mu(t)$ are in turn defined in terms of a basis of $n_\alpha + 1$ cubic B-splines, $\{b_\mu(t)\}_{\mu=0}^{n_\alpha}$, which are constructed over the period $[t_{\text{sd}}, t_{\text{end}}]$ where $t_{\text{sd}}$ and $t_{\text{end}}$ are the start time of social distancing and the end time of the time series respectively (see [65]). Explicitly, we define

$$c_\mu(t) = \begin{cases} b_\mu(t) - \frac{b_\mu(t_{\text{sd}}) b_0(t)}{b_0(t_{\text{sd}})} & \text{if } t \geq t_{\text{sd}} \\ 0 & \text{otherwise.} \end{cases} \tag{7}$$

This definition ensures that: i) $R_i^{\text{res}}(t) = R_0$ for $t < t_{\text{sd}}$ and ii) $R_i^{\text{res}}(t)$ is a continuous function of time regardless of $\alpha_{i,\mu}$ and $b_\mu(t)$ (as can be seen by sending $t \to t_{\text{sd}}$ from above in Eq 7). We fix the start of social distancing as $t_{\text{sd}} = 7$, corresponding to the week beginning March 1 2020 (the first week after the discovery of community transmission) and, based on preliminary investigation, used a basis of $n_\alpha = 8$ functions. Commuters experience a reduction in contact rates through social distancing in both their resident and commuting location (see Eq 4).

In what follows, we use hospitalizations to exclusively refer to those under 60, i.e. the subset we are fitting to. We assumed that the number of weekly infections in each location that are under 60 and require hospitalization, $H_{i,t}$, followed an overdispersed normal distribution [64], with probability density function $f(\mu_{i,t}, \sigma_{i,t}^2)$ specified in terms of the mean and variance. Due to the typical delay between infection and hospitalization, infections were lagged by one week in our hospitalizations model. The mean and variance in the number of weekly hospitalizations in location $i$ at time $t$ are given by,

$$\mu_{i,t} = \rho I_{i,t-1}, \tag{8}$$

$$\sigma_{i,t}^2 = \mu_{i,t}(1 - \rho + \psi^2 I_{i,t-1}), \tag{9}$$

with $\rho$ the probability a case in under 60 and hospitalized and $\psi$ the overdispersion parameter. Together, Eqs 1–9 define a discrete time Markov process with state $\mathbf{X}_t = \{S_{i,t}, I_{i,t}, H_{i,t}\}_{i=1}^{n_c}$ at time $t$.

## Statistical inference

Statistical inference was performed on $T = 56$ weeks of hospitalizations data from $n_c = 5$ counties, $\{H_{i,t}\}_{i=1;t=1}^{n_c;T}$. The likelihood function for the observed data given the model parameters $\Theta$, denoted $\ell(\Theta)$, is found by marginalizing the probability of a trajectory $\{\mathbf{X}_t\}_{t=1}^T$ over the unobserved state variables,

$$
\begin{aligned}
\ell(\Theta) &= \sum_{\mathbf{I}_1, \mathbf{S}_1} \cdots \sum_{\mathbf{I}_T, \mathbf{S}_T} \prod_{t=1}^{T} \left( \prod_{i=1}^{n_c} P(H_{i,t} | I_{i,t-1}, \Theta) \right) P(\mathbf{I}_t, \mathbf{S}_t | \mathbf{I}_{t-1}, \mathbf{S}_{t-1}, \Theta) \\
&= \sum_{\mathbf{I}_1, \mathbf{S}_1} \cdots \sum_{\mathbf{I}_T, \mathbf{S}_T} \prod_{t=1}^{T} \left( \prod_{i=1}^{n_c} f(H_{i,t}; \mu_{i,t}, \sigma_{i,t}^2) \right) P(\mathbf{I}_t, \mathbf{S}_t | \mathbf{I}_{t-1}, \mathbf{S}_{t-1}, \Theta) \\
&= \mathbb{E}\left[ \prod_{t=1}^{T} \prod_{i=1}^{n_c} f(H_{i,t}; \mu_{i,t}, \sigma_{i,t}^2) \Bigg| \mathbf{I}_0, \mathbf{S}_0, \Theta \right],
\end{aligned}
\tag{10}
$$

where $f(H_{i,t}; \mu_{i,t}, \sigma_{i,t}^2)$ is the probability density function of the normal distribution specified by Eqs 8 and 9. If the populations are disconnected ($\epsilon = 0$) then the transmission processes in each location are mutually independent and the likelihood function factorizes, $\ell(\Theta) = \prod_{i=1}^{n_c} \ell_i(\Theta)$, with the likelihood in location $i$ given by

$$\ell_i(\Theta) = \mathbb{E}\left[\prod_{t=1}^{T} f(H_{i,t}, \mu_{i,t}, \sigma_{i,t}^2)\middle| I_{i,0}, S_{i,0}, \Theta\right]. \tag{11}$$

The structure of our model makes Eqs 10 and 11 analytically intractable, and solving either of them requires using Monte Carlo methods, i.e. repeatedly simulating from the transmission model Eqs 1–7. To perform this sampling efficiently we used particle filtering, see [33, 66].

Due to a lack of compelling evidence for pre-2020 immunity to SARS-CoV-2 infection, we assumed the population was initially fully susceptible with one infectious individual in Snohomish county in the week of January 12 (the week and location of the first identified case in Washington state). This infection was not necessarily the ancestor of all subsequent infections as our model allowed for introductions in the force of infection (see Eq 3).

Statistical inference to find maximum likelihood estimates for unknown model parameters was performed using data from five counties in the Puget Sound region: King, Pierce, Snohomish, Thurston and Kitsap. In total, our model had 44 unknown parameters: 4 region-wide parameters and $8 \times 5$ county-specific basis function coefficients. Given the high-dimensionality of the search space, we used a hybrid scheme to maximise the likelihood.

Maximisation over the four region-wide parameters was performed via a grid search. At each grid point, the region-wide parameters were set to their grid point values and the likelihood was maximised over the remaining 40 basis function parameters. To ensure convergence we replicated the estimation 27 times with different initial parameter guesses. Initial guesses for each basis function coefficient $\alpha_{i,\mu}$ within each replicate were generated using latin-hypercube sampling over the interval $[-3, 3]$ (substantially larger than any final estimated values, based on preliminary studies). Likelihood maximisation was then achieved in two stages, both using the iterated filtering algorithm implemented in the mif2 function of the R package "pomp" [32, 33]:

1. To refine our initial guesses for $\{\alpha_{i,\mu}\}_{\mu=1}^{n_\alpha}$, iterated filtering was run for each county separately, assuming no spatial connectivity (i.e. $\epsilon = 0$, see Eq 11). For this first run we used the following iterated filtering hyperparameters: 2000 particles, 50 iterations, a cooling fraction of 0.8 and a random walk standard deviation of $\sigma_{\text{RW}} = 0.05$.

2. The estimates from the first run were used to initialize for a second iterated filtering search with the full connected model (i.e. with $\epsilon$ at its grid point value, Eq 10), which used the same hyperparameters apart from a smaller random walk $\sigma_{\text{RW}} = 0.01$. This second search returned estimates of the basis function coefficients which maximise the likelihood at each grid point.

Using these estimated basis function coefficients, we then calculated the maximimum likelihood at each grid point via particle filtering, again using 2000 particles. Finally, the MLE for all model parameters we found by maximising the resulting likelihood surface over the entire grid. Confidence intervals on the global parameters were constructed in a similar way: For each value of the parameter of interest, the profile likelihood was calculated by maximising the likelihood surface over all the other grid dimensions. Statistical significance was then quantified using a likelihood-ratio test and the application of Wilks' theorem, with results shown at the 95% confidence level.

## Supporting information

**S1 Fig. Time series of weekly confirmed SARS-CoV-2 cases per $10^5$ by Washington state county (Adams–Lewis).**
(TIFF)

**S2 Fig. Time series of weekly confirmed SARS-CoV-2 cases per $10^5$ by Washington state county (Lincoln–Yakima).**
(TIFF)

**S3 Fig. Silhouette score for the agglomerative clustering analysis.**
(TIFF)

**S4 Fig. Repeat of the agglomerative clustering analysis using time series data of weekly COVID-19 hospitalizations.** Maps makes use of TIGER/Line Shapefiles from the U.S. Census Bureau which are in the public doman (https://www.census.gov/geographies/mapping-files.html).
(TIFF)

**S5 Fig. Silhouette score for the agglomerative clustering analysis using hospitalizations data.**
(TIFF)

**S6 Fig. Analysis of age-structured epidemiological data.** A) Weekly confirmed cases by age group B) Weekly age-specific confirmed cases relative to cases aged 20–39 y.o. C) Weekly new hospitalisations by age group D) Weekly age-specific new hospitalizations relative to hospitalizations aged 20–39 y.o.
(TIFF)

**S7 Fig. Heat map of the likelihood surface after maximising over the basis function parameters.** The local maximum in each panel is indicated by a black dot, with the global maximum highlighted in yellow.
(TIFF)

**S8 Fig. Heat map of the likelihood surface found by performing the inference using different initial conditions.** Instead of initialising the outbreak with one infection in Snohomish county in the week beginning January 12, the outbreak simulations were initialised with one infection in King county in the week of January 26 2020. The local maximum in each panel is indicated by a black dot, with the global maximum highlighted in yellow.
(TIFF)

**S9 Fig. Reproduction of Fig 3 using the likelihood surface in S8 Fig.**
(TIFF)

**S10 Fig. Early epidemic take-off in Clark, Spokane and Thurston counties.**
(TIFF)

**S11 Fig. Comparison of simulated trajectories with observed data using a later initial introduction time.** As with the results shown in S8 and S9 Figs, the outbreak was initialised with one infection in King county in the week of January 26 2020. Parameters used were those that maximised the log-likelihood (shown in S8 Fig) subject to $\eta = 0$, i.e. ensuring that the outbreak was seeded from a single introduction.
(TIFF)

## Author Contributions

**Conceptualization:** Tobias S. Brett, Shweta Bansal, Pejman Rohani.

**Data curation:** Tobias S. Brett.

**Funding acquisition:** Shweta Bansal, Pejman Rohani.

**Investigation:** Tobias S. Brett.

**Methodology:** Tobias S. Brett, Pejman Rohani.

**Project administration:** Shweta Bansal, Pejman Rohani.

**Software:** Tobias S. Brett.

**Supervision:** Shweta Bansal, Pejman Rohani.

**Validation:** Tobias S. Brett.

**Visualization:** Tobias S. Brett.

**Writing – original draft:** Tobias S. Brett.

**Writing – review & editing:** Tobias S. Brett, Shweta Bansal, Pejman Rohani.

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
