## [Decision Letter · Decision Letter 0]

12 Oct 2022

Dear Dr Brett,

The reviewers raise a set of points, each of which is individually addressable, but which collectively bring the paper into difficult territory for revisions. At a high level, R1 asks (A) what new insights we gain from this work (while acknowledging that model-fitting is laudable in and of itself);  R2 asks (B) why the county spatial scale is a good one, given that the major conclusion of the paper is that a statewide aggregated scale is too large; And R1 and R3 (C) both observe that parameters like epsilon and the two components of rho are surely time-varying (and whose time-varying values could thus explain more of the variation in the observed data). 

These points, as well as the others (e.g. using pre-pandemic mobility data), may be difficult to collectively address in a way that leads to interpretable conclusions. One would hope that the modeling results are sufficiently reliable as to be interpretable, but points B and C call this into question, while point A questions the value of the insights regardless.

My consensus choice is Major Revision, but I also recognize that you may feel that addressing the reviewers' points may be too much effort. If you do choose to revise the paper, please don't hesitate to contact the journal if additional time is required. If you prefer to transfer the paper toward PLOS One, please also let me know.

Thank you very much for submitting your manuscript "Charting the spatial dynamics of early SARS-CoV-2 transmission in Washington state" for consideration at PLOS Computational Biology.

As with all papers reviewed by the journal, your manuscript was reviewed by members of the editorial board and by several independent reviewers. In light of the reviews (below this email), we would like to invite the resubmission of a significantly-revised version that takes into account the reviewers' comments.

We cannot make any decision about publication until we have seen the revised manuscript and your response to the reviewers' comments. Your revised manuscript is also likely to be sent to reviewers for further evaluation.

Sincerely,

Daniel B Larremore, Ph.D.

Academic Editor

PLOS Computational Biology

Virginia Pitzer

Section Editor

PLOS Computational Biology

Reviewer's Responses to Questions

**Comments to the Authors:**

Reviewer #1: The manuscript by Brett et al. describes a model for SARS-CoV-2 transmission fitted to data from Washington State. Some of the major conclusions drawn are about the role of stochasticity and mobility at different phases of the epidemic there.

Overall, the model was nicely done and its fit to data was good. Technically speaking, this is no small feat. With respect to the results and conclusions drawn, I think the authors more or less did what they could but that the insights gained, combined with the rigor of those insights, were more modest. There are a few respects described below in which the structure of the model was a limitation in how convincingly the authors could draw a certain conclusion. The impact of this work would be greater if more flexible or perhaps multiple models were used. There is also the issue that these results pertain only to one state and that none of the results are particularly surprising at this stage in the pandemic, given all the modeling work that has been published.

l 20 - It's a bit unconventional to refer to figures from the paper that present data in the introduction.

l 52 - Is there anything else this pattern could be "consistent with"? For example, the timing of case reporting.

l 55 - This seems like a higher estimate of the probability of detecting infections than I have generally seen, especially for early in the pandemic.

l 319 - The analysis focused on data on hospitalizations to ameliorate changes over time in reporting probability. At the same time, there was nothing in the model to help deal with the age of cases, which changed over the course of the pandemic and affected the ratio of hospitalized cases to infections.

l 325 - The model relies on pre-pandemic mobility data from commuting patterns. Given the centrality of mobility to the results and the drastic changes that occurred to mobility early in the pandemic, this is a major concern about the study.

methods - The model and inference procedure are sound and well-done.

methods - However, it is a bit disappointing that only 5 counties can be handled at once, presumably due to the dimensionality of the parameter space. It would be helpful if the authors could state that explicitly if so.

methods - I also did not see how uncertainty on model parameters was quantified given the maximum-likelihood approach.

l 130 - Modify "above" to "in the methods" since the methods is located at the end of the manuscript.

l 166 - While the atuhors' investigation of what drove the estimate of epsilon is useful, it eschews the larger issue of the fact that mobility varied substantially during this period. While they account for that with time varying reproduction number, it certainly should have resulted in a time varying epislon, too. This highlights a serious constraint of the model in that it fixes epsilon over time.

l 167 - While the stuff about Qt is interesting, it could be stressed more that this analysis only gives a vague, qualitative feel for the impact of stochasticity early in the pandemic, rather than a clearly interpretable parameter estimate or something (e.g., if there were a parameter in the model controlling the degree of heterogenity in transmission).

l 212 - I wonder if the bit about early superspreading could be misinterpreted by readers. The model did not have any structure to handle superspreading per se, so this is really about stochasticity. The thing that I think is potentially misleading about how this is presented is that it could be interpreted to imply that superspreading is a general property of SARS-CoV-2 transmission that was also in effect later in the pandemic. While the reality is that that probably is the case, I do not see how this study provides support for an assertion like that. As such, I think the authors should be much more careful in how the present the conclusions drawn on this matter.

l 217 - In this paragraphs, the authors seem to want to have it both ways. On the one hand, aggregating at the state level is too much. On the other hand, things were pretty synchronized and similar.

l 304 - during during

Reviewer #2: In the study, authors used two statistical methods to examine the spatial transmission pattern of SARS-COV-2 in Washington State during the 1st and 2nd wave (prior to mass vaccination/ VOCs). The first method (clustering analysis) identified several spatial clusters and the 2nd used a metapopulation spatial model along with the IF to estimate key parameters and infer underlying transmission dynamics for a subset of the WA counties at that time. The paper addresses important questions and reads interesting. However, it has some major errors/problems that require a reanalysis and likely revision of their main findings and conclusions.

Major issues:

1. The foremost problem with the paper is that, based on earlier case reporting/studies, authors assumed that SARS-COV-2 was introduced into WA in mid-January 2020 and continued to spread afterwards. This assumption was used to formulate the initial model conditions and substantially determined their model results and the conclusion that “super-spreading” has to occur to have the observed dynamics (i.e. little transmission prior to March 2020, followed by explosive expansion of infection). However, a later study in Worobey et al. 2020 Science (https://www.science.org/doi/10.1126/science.abc8169) showed that, while SARS-COV-2 was initially introduced in mid-Jan 2020, that initial spread was eliminated due to extensive public health intervention at that time; the later pandemic wave was caused by subsequent reintroduction later on, rather than from the presumed silent, continued transmission since mid-Jan 2020. Because of this major factual error/assumption and its impact on model inference findings, authors should redo their 2nd analysis per findings from Worobey et al.

2. Authors used hospitalization data (for ages under 60) for model inference (2nd analysis). In the model, the parameter rho (probability of an infection is under 60 and hospitalized) was used to link model-estimated infection to the under 60 hospitalization data. Rho is a combination of two probabilities, both could change over time. First, the probability of an infection being under 60 could change and likely did change substantially over time due to changes in infection demographics, as different interventions affected different population segments over time. The 2nd probability, hospitalization given infection, also changed over time; particularly, with better case management and outpatient treatment, hospitalization rate decreased in later months (likely after the initial wave, but I don’t have a good reference for this; so please check the literature). So in combination, I would expect rho to vary, not so trivially, over time; and as such, treating rho as a constant parameter (even though it is estimated per the data) would be problematic. Looking at Fig 7, the estimated cumulative infection rates were fairly low. Authors overlapped their estimates with the CDC serology data from Sep 2020 onward to show their estimates were in line with the serology data. But serology data have issues due to sero-revision. So I wonder if the model underestimated the infection rate due to the use of the constant rho. At least some sensitivity analysis is needed to address this.

Other comments:

3. 1st paragraph of Introduction: i) SARS-COV-2 was detected in Dec 2019, not Jan 2020; ii) problem with the description of the initial introduction, as noted in comment #1; iii) omega should be Omicron.

4. Analysis 1. Only case time series were used for the clustering analysis. Have authors tried using hospitalization and mortality data as well? And how consistent are the results, given all those data have some issues/biases.

5. Page 7, the very large drop in R_i^res in Thurston and Kitsap. Looking at Fig 4, there were much fewer hospitalizations in those counties. How much is this estimate due to the much lower level of early transmission in those counties, rather than a drop due to lockdown?

6. Page 9 and Fig 5. I don’t understand what the figure is showing, especially the results on Qt and Fig 5D. Is Qt cumulative or for each week? If Qt is cumulative, it does not look like there is much difference after March 2020. Could authors please clarify?

7. Hospitalization data have issues that warrant some discussion on study limitation. Before March 2020, there was limited testing, leading to under counting. It could also be affected by healthcare seeking behavior, e.g., awareness and concern of covid could drive more people to seek treatment. These systemic biases/similar behavior across population that affected the data could have contributed to the apparent synchrony.

Reviewer #3: This is quite an interesting paper looking at whether / why / how we can reconstruct effective connectivity matrices between spatial regions according to the reported case counts at that scale. I think the paper is quite well written, the figures are clear, and the approach itself is interesting. While I have a tough time imagining this technique being deployed for nearly real-time policy response (it would seem to require more data to validate if one were applying this technique in real-time), it can be used to retrospectively characterize the often hidden connectivity between regions. I think that'a great! There are a couple small things I have questions about, but overall, I think this is a really nice paper that doesn't require too many tweaks to see it through.

1. I have a few broad questions about robustness of these measures to spatial/temporal up/down-scaling, which I think could improve this method's usefulness in broader contexts outside of Washington. The reason for this question is that I'm imagining what it would look like to apply it in other contexts, say, Arizona and Massachusetts (I'm not asking for this exact analysis, just as examples). Both of these states have ~15 counties, but Massachusetts reports much of its case counts / hospitalization data at the *municipality* level in addition to counties (n=351 vs n=15). Municipalities in MA are largely subsumed by counties, so in theory, one could start from a correlation matrix as in Fig 2A for both the county-level and municipality-level correlations. The reason I'm curious about these different spatial scales is because of the authors' discussion and analyses relating to super-spreading in the early months of the pandemic. The authors' explanation is largely compelling, but would that same conclusion be drawn if only we had more spatially granular data of case counts?

More generally, it would be helpful to know how the results depend (or don't) on the selection of spatial scale that cases are reported at. One approach would be to artificially merge Washington counties into "double counties" and re-run the analyses as if Washington had half the number of counties but ones that were larger in size and population than they currently are. You can do this again and again until all the counties form a single mega-county that subsumes the whole state --- in that case, the reported cases time series at the state level would be perfectly synchronized (trivially so, as it is only being compared with itself), whereas one can imagine the spatial correlation between cases at the arbitrarily small scale (e.g. partitioning the state of Washington into tons of 20x20 meter parcels) may not be as informative as the scale used in this paper.

I would like to see the authors contend with this a bit more, addressing and attempting to quantify how exactly the effectiveness of this approach depends on the spatial scale of the reported data and the temporal scale at which it is reported (i.e. the same discussion above is still relevant when imagining up-sampled or down-sampled time series data; how would this approach change if data were only reported at 2-week intervals? 1-week? Daily?).

2. Stepping back, it also seems like the authors could use this to provide stronger / more normative explanations about data reporting procedures nationwide. That is, if this approach can prove to be a useful stand-in or complement to the large scale mobility datasets that some research groups have access to, what is the best / most appropriate spatial scale to report data at to ensure that the spatial granularity is maximally informative to the virus that's spreading?

3. Lastly, I really think the idea of a "resident reproductive number", and I think its use here is important for future work looking at the potentially vast spatial heterogeneities that may exist depending on whether the R_res is calculated at the census tract level, county, state, etc.

4. Very small: add a "by" to line 275 -- "may have been facilitated ___ the mobility patterns"

**Have the authors made all data and (if applicable) computational code underlying the findings in their manuscript fully available?**

Reviewer #1: **No: **They say they will do it later. A github repository while in review and then zenodo later would be better. Something like that should really be journal policy.

Reviewer #2: **No: **I don't see any mentioning about computational code

Reviewer #3: Yes

PLOS authors have the option to publish the peer review history of their article (what does this mean?). If published, this will include your full peer review and any attached files.

Reviewer #1: No

Reviewer #2: No

Reviewer #3: No
---

## [Decision Letter · Decision Letter 1]

12 May 2023

Dear Dr Brett,

Thank you very much for submitting your manuscript "Charting the spatial dynamics of early SARS-CoV-2 transmission in Washington state" for consideration at PLOS Computational Biology. As with all papers reviewed by the journal, your manuscript was reviewed by members of the editorial board and by several independent reviewers. The reviewers appreciated the attention to an important topic. Based on the reviews, we are likely to accept this manuscript for publication, providing that you modify the manuscript according to the review recommendations.

As you can see, 2/3 reviewers were satisfied, yet the third raises a number of questions about the study's conclusions, claims, and caveats. I ask that the authors please address these remaining concerns with changes to the text of the paper, taking care to avoid any claims that contradict themselves (a concern of the reviewer) or claims that are too bold given data limitations (another concern).

Sincerely,

Daniel B Larremore, Ph.D.

Academic Editor

PLOS Computational Biology

Virginia Pitzer

Section Editor

PLOS Computational Biology

As you can see, 2/3 reviewers were satisfied, yet the third raises a number of questions about the study's conclusions, claims, and caveats. I ask that the authors please address these remaining concerns with changes to the text of the paper, taking care to avoid any claims that contradict themselves (a concern of the reviewer) or claims that are too bold given data limitations (another concern).

Reviewer's Responses to Questions

**Comments to the Authors:**

Reviewer #1: I am satisfied with the authors' extensive revisions.

Reviewer #2: I appreciate the authors revise the paper to address my comments. There are a few things I still don’t agree/understand:

1) Worobey et al. showed that there was at least one additional introduction in early February that likely led to the subsequent pandemic wave in WA. I don’t think they precluded additional introductions or multiple introductions around that time. And given that SARS-COV-2 had expanded globally by Feb/March 2020, with multiple outbreaks in Europe and silent spread in places like NYC, multiple introductions around Feb 2020 was more likely than not. So I don’t understand the authors’ argument that b/c eta was estimated to be ~8, it was impossible to have a later start of the pandemic wave in Feb 2020.

Relatedly, eta (the seeding) is probably time-varying. So it’s probably not appropriate to treat it as a constant. Similar problems exist for other time-varying parameters (rho etc.). So in general, I think it is somewhat problematic to use the IF, which is designed for time-stationary parameter estimates, for the study here. At the least, I’d hope the authors use larger perturbations in the IF to add the necessary uncertainty to those time-varying parameters.

2) I don’t understand what Fig S6 is showing nor the authors’ reply to my comment related to the changing hospitalization-to-infection ratio. The data show hospitalizations (i.e. the nominator) yet the denominator –i.e., infections – is not observed and infection demographics likely changed substantially, due to changing NPIs and mobility. Unless the model is age-specific and uses age-specific hospitalization-to-infection ratio, the non-age-specific hospitalization ratio likely would vary with the infection demographics and it probably would not be so trivial.

3) Authors commented on hospitalization data being too noisy for their 1st analysis (clustering analysis) yet used hospitalization data for the 2nd analysis. That sounds contradictory.

4) “We see the largest rise in Qt during March and April, implying that the impact of early stochastic events persisted over the entire period of study. Subsequently, there is little increase in Qt between April and October 2020, implying that individual stochastic fluctuations have limited lasting impact.” I don’t understand this take. If there was not much change in Qt after April 2020, “implying that individual stochastic fluctuations have limited lasting impact” (2nd part of the sentence), why the first part says “the impact of early stochastic events persisted over the entire period of study”?

5) The closing sentence says “Finally, our results demonstrate that accurate prediction of future spread within densely populated urban areas during periods of low incidence is not feasible without: i) modelling approaches capable of accounting for stochastic transmission events and ii) access to zip-code- or neighbourhood-level epidemiological and mobility data, at a minimum.” This is somewhat contradicting. If it is mostly stochastic, I don’t think it is predictable, no?

6) Overall, I think there is substantial uncertainty (e.g., noted in my comments above) and simplification (some noted by the authors themselves)/likely misspecification in the study. Given the uncertainty, I would recommend the authors at least tone down their statements on the findings (e.g., sentences in the concluding paragraph).

Reviewer #3: The authors have conducted quite an extensive, impressive series of edits, which have satisfied my original questions and concerns—especially those related to the most-appropriate spatial scale for data in this procedure. I'd like to commend their efforts and note that I am satisfied with their revision.

**Have the authors made all data and (if applicable) computational code underlying the findings in their manuscript fully available?**

Reviewer #1: None

Reviewer #2: None

Reviewer #3: Yes

PLOS authors have the option to publish the peer review history of their article (what does this mean?). If published, this will include your full peer review and any attached files.

Reviewer #1: No

Reviewer #2: No

Reviewer #3: No

Figure Files:

Data Requirements:

Reproducibility:

References:

---

## [Editor Report · Decision Letter 2]

12 Jun 2023

Dear Dr Brett,

We are pleased to inform you that your manuscript 'Charting the spatial dynamics of early SARS-CoV-2 transmission in Washington state' has been provisionally accepted for publication in PLOS Computational Biology.

Best regards,

Daniel B Larremore, Ph.D.

Academic Editor

PLOS Computational Biology

Virginia Pitzer

Section Editor

PLOS Computational Biology

---

## [Editor Report · Acceptance letter]

26 Jun 2023

PCOMPBIOL-D-22-01274R2 

Charting the spatial dynamics of early SARS-CoV-2 transmission in Washington state

Dear Dr Brett,

I am pleased to inform you that your manuscript has been formally accepted for publication in PLOS Computational Biology. Your manuscript is now with our production department and you will be notified of the publication date in due course.

With kind regards,

Anita Estes
